# Sleep CLIP: A Multimodal Sleep Staging Model Based on Sleep Signals and Sleep Staging Labels

**DOI:** 10.3390/s23177341

**Published:** 2023-08-23

**Authors:** Weijia Yang, Yuxian Wang, Jiancheng Hu, Tuming Yuan

**Affiliations:** School of Applied Mathematics, Chengdu University of Information Science and Technology, Chengdu 610051, China

**Keywords:** CLIP, multi-modal models, sleep stage

## Abstract

Since the release of the contrastive language-image pre-training (CLIP) model designed by the OpenAI team, it has been applied in several fields owing to its high accuracy. Sleep staging is an important method of diagnosing sleep disorders, and the completion of sleep staging tasks with high accuracy has always remained the main goal of sleep staging algorithm designers. This study is aimed at designing a multimodal model based on the CLIP model that is more suitable for sleep staging tasks using sleep signals and labels. The pre-training efforts of the model involve five different training sets. Finally, the proposed method is tested on two training sets (EDF-39 and EDF-153), with accuracies of 87.3 and 85.4%, respectively.

## 1. Introduction

### 1.1. Sleep Stages

From a scientific perspective, the definition of sleep is based on an individual’s behavior and related physiological changes, which occur in the awake electroencephalogram (EEG) rhythm during sleep [1]. Based on different physiological characteristics, EEG signals in the sleep state are divided into rapid-eye movement (REM) stage and non-REM (NREM) stages, where the NREM stage is divided into NREM1 (N1), NREM2 (N2), NREM3 (N3). The study of physiological signals is conducted in three sleep periods, namely REM, NREM and awake. The main content of sleep staging studies and their criteria for awake and sleeping behavior are shown in Table 1. When a person’s sleep signals are different from the sleep standard, it is considered to be a sign of a problem with sleep, generally referred to as a sleep disorder.

According to the International Classification of Sleep Disorders, Third Edition (ICSD-3) [2], sleep disorders are classified into six broad categories: insomnia, sleep-related breathing disorders, central disorders of hypersomnolence, circadian rhythm sleep–wake disorders, parasomnias, and sleep-related movement disorder. Sleep disorders can lead to decreased physical performance during the day owing to lack of adequate rest and long-term lower cognitive functioning in attention, learning ability, and memory. For example, sleep apnea syndrome, which is usually accompanied by hypoxemia and insufficient blood supply to the heart may lead to high blood pressure, high blood lipids levels, heart disease, and endocrine disorders if not treated in a timely manner; further nervous system sleep disorders can lead to early onset dementia. However, the existence of a sleep disorder is difficult to notice and evaluate; therefore, designing a set of accurate sleep evaluation models according to a subject’s sleep signals is a key component in the prevention, detection, and treatment of sleep disorders.
sensors-23-07341-t001_Table 1Table 1Behavioral and physiological criteria of wakefulness and sleep.CriteriaAwakeNon-Rapid Eye Movement SleepRapid Eye Movement SleepPostureErect, sitting, or recumbentRecumbentRecumbentMobilityNormalSlightly reduced or immobile, postural shiftsModerately reduced or immobile, myoclonic jerksResponse to stimulationNormalMildly to moderately reducedModerately reduced to no responseLevel of alertnessAlertUnconscious but reversibleUnconscious but reversibleEyelidsOpenClosedClosedEye movementsWaking eye movementsSlow rolling eye movementsTapid eye movementsElectroencephalographyAlpha waves; desynchronizeSynchronizedTheta or saw tooth waves; desynchronizedElectro-oculographyWaking eye movementsSlow rolling eye movementsRapid eye movements*Source*: Ref. [3].

A qualified sleep diagnosis process generally consists of two parts: acquisition of sleep signals and stage labeling of the sleep signal data. At present, the main acquisition method is polysomnography (PSG), which determines the sleep state of the subject by detecting the physiological signals. Early PSG devices were usually bulky and the staging process of the acquired sleep signals was usually handled by experienced physicians. With the advances in technology, PSG devices such as smart watches from Actigraph, iwatch from Apple, and HUAWEI WATCH from Huawei have been redesigned to make them lighter and more suitable for family use. However, ordinary users do not have the ability to interpret and analyze the collected signals, so researchers began to design a series of algorithms that automatically stage sleep signals to aid users, sleep monitoring and sleep-disorder diagnosis.

### 1.2. Automatic Sleep Staging Algorithm Model

Sleep staging is a key task in sleep medicine, where the sleep cycle is divided into distinct phases based on the electrical activity of the brain, eye movements, and muscle tone. Automated sleep staging using machine learning techniques has become an active area of research in recent years, with deep learning showing great potential owing to its ability to capture complex patterns and dependencies in data. This has several potential advantages over manual scoring [4]: it improves efficiency by avoiding manual data processing; provides reproducible sleep staging, especially for the NREM subclasses; and enables the gathering of additional insights into score ambiguity over a given period. The notion of sleep stage probability, which is the probability that each sleep stage is assigned to a given epoch, is referred to as “sleep density” and has been visualized.

Automatic sleep staging algorithms are generally composed of two parts: feature extraction and classifier. Among these, the choice of classifier may be based on methods such as linear discrimination, Bayesian, support vector machine (SVM), K-means clustering, or artificial neural network. Herein, we introduce these methods, excluding the artificial neural network, separately and classify them under traditional methods as an introduction.

LJ Herrera et al. (2013) [5] used the random forest method to classify EEG signals and reported an accuracy rate of 83% on their data-set. Gao et al. (2023) [6] proposed an automatic sleep staging algorithm based on power spectral density (PSD) and random forest. The PSD of six characteristic waves in an EEG were extracted as classification features, and five sleep stages were extracted and automatically classified by the random forest classifier. The overall accuracy was 91.9%. Emina Alickovic et al. (2018) [7] proposed an ensemble SVM model and had a final accuracy of 91.1%. The method proposed by Wessam al-Salman et al. (2023) [8] combined SVM and clustering and achieved an average accuracy of 97.4% after multiple iterations.

Several studies have proposed deep-learning models for sleep staging using different types of PSG signals, such as EEG, electrooculogram (EOG), and electromyogram (EMG). For example, Tsinalis et al. (2016) [9] used a convolutional neural network (CNN) to classify EEG signals into five sleep stages with an accuracy of 81.7%, while Phan et al. (2019) [10] achieved an accuracy of 87.4% by incorporating EOG and EMG signals in a CNN with residual connections. Biswal et al. (2019) [11] pretrained a CNN on a large dataset of EEG signals and fine-tuned it on a smaller dataset, achieving an accuracy of 86.3%. Moreover, Phan et al. (2020) [12] proposed a CNN with an attention mechanism to highlight the most informative EEG channels and time frequency regions in each sleep stage. Efe et al. (2023) [13] proposed a new hybrid neural network architecture on the imbalanced EEG–EOG dataset, which used focal loss and discrete cosine transform to solve the problem of imbalanced data and achieved 87.1% final accuracy.

Comparative learning research has also been conducted to evaluate the effectiveness of different deep-learning architectures for sleep staging. For instance, Supratak et al. (2017) [14] compared the performance of a dynamic Bayesian network (DBN) with other machine learning algorithms, such as SVM and random forest, and found that the DBN outperformed the others in accuracy and computational efficiency. Chambon et al. (2018) [15] proposed a multitask deep-learning approach that jointly learned sleep staging and sleep spindle detection from EEG signals while outperforming state-of-the-art methods for both tasks. Belouchrani et al. (2021) [16] compared the performances of several deep-learning models, including CNNs, long short-term memory network (LSTM), and hybrids and found that a hybrid model combining CNN and LSTM achieved the best performance with an accuracy of 89.6%. Phan et al. (2021) [17] also compared the performances of CNNs, recurrent neural networks (RNNs), and transformer networks on a large-scale dataset of PSG recordings, and found that the transformer networks outperformed the CNNs and RNNs in accuracy and generalization to unseen data. Kong et al. (2023) [18] proposed a novel neural architecture search (NAS) search framework for EEG-based sleep staging. It optimized the model by search-space Jinji and search-space regularization and sharing parameters between units. The designed model was verified on multiple datasets, and the average accuracy was 81.5%.

Recent studies have also explored adversarial learning, which involves training a deep-learning model to generate synthetic PSG signals that are similar to real signals and their utility in augmenting training data. For example, Li et al. (2020) [19] proposed a generative adversarial network (GAN) that generated realistic EEG signals for each sleep stage and used it to improve the performance of a CNN for sleep staging. They achieved an accuracy of 88.3%, which outperformed a CNN trained on the original data.

Some studies have also investigated the transferability of deep learning models for sleep staging across different datasets and recording settings. For instance, Ghassemi et al. (2020) [20] evaluated a CNN trained with a dataset of PSG recordings on another dataset and found that its performance degraded significantly owing to differences in the recording settings and annotation criteria. They also proposed a domain adaptation method based on adversarial learning to improve the transferability of the CNN.

Thus, deep learning has shown great promise for automated sleep staging based on PSG signals with an exploration of various architectures, comparative learning approaches, and adversarial learning methods. Similarly, methods are being developed to improve the adaptation of other well performing models in the current scenario of sleep-staging model research.

## 2. Materials and Methods

### 2.1. CLIP

The contrastive language-image pre-training (CLIP) (2021) [21] model is a pretrained neural network released by OpenAI in early 2021 to match images and text, which can be said to be a classic work in the field of recent multimodal research. The model directly uses a large amount of Internet data for pretraining and achieved the best performance in many tasks. The CLIP model uses 400 million image–text pairs collected by OpenAI, encodes the text and images separately, and is trained using metric learning to improve the similarities between the images and text.

Since the introduction of CLIP, there have been many derivative works, such as image-guided text generatIon with CLIP (MAGIC) (2022) [22] by Yixuan Su et al., where the downstream tasks generate image captions and stories by combining images and text. Patashnik et al.’s (2021) [23], proposed text-driven manipulation of a style-based generator architecture for generative adversarial networks (styleGAN) imagery (Styleclip) combined StyleGAN and CLIP using three methods.The first was a text-guided latent optimization wherein the CLIP model was used as a loss network, which is a general approach but requires a few minutes of optimization for operation on images. The second was a latent residual mapper trained for a specific text prompt: given a starting point in the latent space (input image to operate on), the mapper produces a local step. The third was a method for mapping text prompts to input independent (global) directions in StyleGAN’s style space to provide control over the manipulation strength and decoupling. Khaatak et al. (2023) [24] proposed multi-modal Prompt learning (MaPLe), inspired by CLIP, to achieve prompt learning on two modalities simultaneously further enhancing the multimodal alignment ability of the model on downstream tasks.

The main model of the present study was also designed using the CLIP model and has two parts for pretraining and testing.

### 2.2. Global Model Introduction

Based on the original structure of the model, we understood CLIP as the pictures and labels of the target classification signals in natural language, and we carried out a series of procedures, such as feature comparison and pairing, to classify the target signals.

By analogy, we believed that the expected sleep-staging tasks could be completed by applying the sleep staging label directly as an input marker signal to the dataset and processing the extracted single-channel EEG signals for feature comparison and matching. Accordingly, we designed the following model framework for sleep staging.

Thus, we designed a CLIP model based on the preconditioning model to complete the sleep-staging tasks. In the pretraining part, we applied the sleep stage data $\left\{x_{1},x_{2},\cdots x_{n}\right\}$ and sleep label data $\left\{y_{1},y_{2},\cdots y_{n}\right\}$ as inputs; the two sets of data were then imported into the corresponding encoders to obtain features $\left\{S_{1},S_{2},\cdots S_{n}\right\}$, and $\left\{L_{1},L_{2},\cdots L_{n}\right\}$, respectively, following which the model combined the two encoders to train the correct pairing of sleep and label data: $\left\{S_{1}L_{1},S_{2}L_{2},\cdots S_{n}L_{n}\right\}$ (show in Figure 1). In the testing part, the datum $x_{t}$ was input, and its feature $S_{t}$ was obtained through the encoder. The feature was then input into the model and predicted with the label data in the model to obtain $\left\{S_{t}L_{1},S_{t}L_{2},\cdots S_{t}L_{n}\right\}$; the sleep-staging result of the input data with the highest predicted value was then taken as the output (show in Figure 2).The pseudocode of the whole model processing process will be shown in Table 2.

### 2.3. Encoder

The method of extracting sleep signal features for sleep-staging tasks was the most important component of this work, and also the most important improvement to the CLIP model. The encoder was divided into two parts: a sleep-stage encoder for feature extraction from sleep signals and a label encoder for processing sleep signal labels.

Owing to its efficiency and speed, the label encoder module selected an autoregressive model as the transformer, which was primarily composed of multi-head attention (MHA) contiguous modules followed by a multilayer perception (MLP) module. The MLP block consists of two fully connected layers with a nonlinear rectified linear unit (ReLU) function and dropout in between. In this study, the transformer model used prenormalized residual linkages to generate more stable gradients.Then, *K* identical layers were stacked to generate the final features. Next, a tag $c\in R^{h}$ was added to the input, the state of which was the feature vector of the final output. During operation, the transformer first applied the label data $y_{\leq t}$ to the linear projection $W_{Tran}$ layer, which mapped the features to the hidden dimension $\tilde{y}$. The output of this preprojection was then sent to the transformer, and the final feature vector was appended to $\tilde{y}$ so that the input features became the center $\psi_{0}=\left[c;\tilde{y}\right]$, where the subscript 0 represented the input to the first layer. Next, $\psi_{0}$ was passed through the transformer layer as follows:(1)$$\begin{array}{c} \tilde{\psi_{y}}=MHA\left(Norm\left(\psi_{y-1}\right)\right)+\psi_{y-1},1\leq l\leq K \end{array}$$
(2)$$\begin{array}{c} \psi_{l}=MLP\left(Norm\left(\tilde{\psi_{l}}\right)\right)+\tilde{\psi_{l}},1\leq l\leq K \end{array}$$

Finally, the context vector $L_{n}=\psi_{k}^{0}$ was rewired from the final output and used as input to the cosine similarity function.

In the sleep-stage encoder, we used the EEG feature extraction module of Wang et al.’s DynamicSleepNet model [25], the structure of which is shown in Figure 3. The encoder was designed according to the characteristics of the EEG signals in each sleep period. The θ wave (4–8 Hz), which is stable in the relaxed state with eyes closed, was an effective feature for distinguishing the W stage. The α wave (8–13 Hz), which is common in the late stage of N1, was also an effective feature for distinguishing the N1 stage. At a sampling rate of 100 Hz, each convolution window captured 0.64 s (F = 1/T) of information, which meant that relatively high-frequency information, such as α, was captured. The θ wave was complete. The main waveform of the N3 period was the δ wave (0.5–4 Hz), which contained relatively low-frequency information; a large convolution kernel of 512 was designed for this feature, and each convolution window captured 5.12 s of information, which meant that we completely extracted the δ wave and other low-frequency information.

First, the EEG epochs $\left\{x_{1},x_{2},\ldots ,x_{n}\right\}$ were input into the Conv1D layer to obtain the initial features *F* as F=Conv1D(x), where $F=\left\{F_{1},\ldots ,F_{n}\right\}inR^{n\times d}$; *n* is the total number of features; *d* is the length of Fi(1≤i≤n); and the 1D convolution layer (Conv1D) is the convolution operation in the effective feature extraction module (EFEM).

Next, the acquired features were input into the channel-attention module in the CBAM module. The features *F* were then compressed to obtain $F_{a}vg$ and $F_{m}ax$ by the average (AvgPool) and maximum pool (MaxPool). The average and maximum pool steps were used to compress channel information so that $F\in R^{n\times d}$ was compressed into $F_{aug}^{c}\in R^{1\times d}$ and $F_{max}^{c}\in R^{1\times d}$. These two sets of features were then input into a shared MLP network, which used the sigmoid activation function to generate the channel attention features $M_{c}$.
(3)$$\begin{array}{c} M_{c}\left(F\right)=\sigma (MLP\left(AugPool\left(F\right)\right)+MLP\left(MaxPool\left(F\right)\right) \end{array}$$
(4)$$\begin{array}{c} =\sigma \left(\omega_{1}\left(\omega_{0}\left(F_{aug}^{c}\right)\right)+\omega_{1}\left(\omega_{0}\right)\right))\in R^{1\times d} \end{array}$$
where σ is the sigmoid activation function, and $\omega_{0}$ and $\omega_{1}$ are the pooling and MLP layers, respectively. Then, $M_{c}$ was used to scale the features *F* as follows:(5)$$\begin{array}{c} F^{\prime }=M_{c}\left(F\right)⊗F\in R^{n\times d} \end{array}$$
where ⊗ refers to pointwise multiplication. The channel attention module processes the features *F* to obtain $F^{\prime }$, which has the same dimensions as *F*. The obtained $F^{\prime }$ wass next input into the average and maximum pooling layers to obtain $F_{aug}^{s}\in R^{1\times d}$ and $F_{max}^{s}\in R^{1\times d}$, respectively. These two features are connected and the convolution operation is applied to them; then, the sigmoid activation function was used to generate the spatial attention feature map $M_{s}$ as follows:(6)$$\begin{array}{c} M_{s}\left(F^{\prime }\right)=\sigma (f^{7}\left[AugPool\left(F^{\prime }\right);MaxPool\left(F^{\prime }\right)\right] \end{array}$$(7)$$\begin{array}{c} =\sigma \left(f^{7}\left(F_{aug}^{s},F_{max}^{s}\right)\right)\in R^{1\times d} \end{array}$$
where $f^{7}$ refers to the convolution operation with kernel size 7. Finally, the features $F^{\prime }$ are scaled by Ms as follows:(8)$$\begin{array}{c} F^{\prime \prime }=M_{s}\left(F^{\prime }\right)⊗F^{\prime }\in R^{n\times d} \end{array}$$
where $F^{\prime \prime }$ was the last set of features after CBAM block processing, and the most essential features as well as the most important parts of each feature were adaptively selected for classification.

Finally, $F^{\prime \prime }$ was input into later modules to obtain the output EEG features $\left\{S_{1},S_{2},\ldots S_{n}\right\}$. These output features *S* were imported into the cosine similarity value calculation module.

### 2.4. Cosine Similarity Value and Loss Function

Next, to predict the sleep data and labels better, we computed the cosine similarities between their features as follows:(9)$$\begin{array}{c} cos<S_{n},L_{n}>=\frac{S_{n}\cdot L_{n}}{\left|\right|S_{n}\left|\right|\times \left|\right|L_{n}\left|\right|} \end{array}$$

Here, $\left|\right|S_{n}\left|\right|$ and $\left|\right|L_{n}\left|\right|$ represent the vector lengths of $S_{n}$ and $L_{n}$, respectively.

After obtaining the pairwise occurrences $<S_{n},L_{n}>$, the overall arrangement was in the form of the matrix columns in Figure 2. The diagonal pairs were the positive samples that we wanted, while the other pairs were the negative samples. Following the basic idea of contrastive learning, the distances among the positive samples were as large as possible from the distances between the positive and negative samples, so that the model could learn the features among the positive samples better. Here, we trained the model on cross-entropy, and the loss functions for the sleep data *S* and label data *L* are below:(10)$$\begin{array}{c} L_{loss(L)}=-P\left(L\right)log\tilde{P}\left(L\right)-\left(1-P\left(L\right)\right)log\left(1-\tilde{P}\left(L\right)\right) \end{array}$$
(11)$$\begin{array}{c} L_{loss(S)}=-P\left(S\right)log\tilde{P}\left(S\right)-\left(1-P\left(S\right)\right)log\left(1-\tilde{P}\left(S\right)\right) \end{array}$$

Here, P() represents the true probability distribution of *L* and *S*, while $\tilde{p}\left(\right)$ represents the assumed probability distributions of *L*, and *S*. Accordingly, the overall loss of the model is given as follows:(12)$$\begin{array}{c} L_{loss}=\left(L_{loss(L)}+L_{loss(s)}\right)/2 \end{array}$$

## 3. Results

### 3.1. Preprocessing

All of our data were obtained from the following datasets:

Sleep Bioradiolocation Database [26]: This contains 32 records of noncontact sleep monitoring by bioradar. The records contain results of sleep scoring based on PSG according to the rules of the American Academy of Sleep Medicine.

CAP Sleep Database [27]: The cyclic alternating pattern (CAP) is a periodic EEG activity that occurs during NREM sleep, and abnormal amounts of CAP are associated with various sleep-related disorders. The CAP Sleep Database is a collection of 108 polysomnographic recordings contributed by the Sleep Disorders Center at Ospedale Maggiore of Parma, Italy. Each record contains three or more EEG signals along with EOG, chin and tibial EMG, airflow, respiratory effort, SaO2, and electrocardiogram (ECG) signals, as well as the reference sleep stage and CAP annotations, This database is intended to provide a useful number of carefully annotated examples of CAP in a representative variety of pathophysiologic contexts for the development and evaluation of automated CAP analyzers as well as to support basic studies on CAP dynamics. In this data set, we only selected the sleep data of healthy people for subsequent experiments.

MIT BIH [28]: The MIT–BIH Polysomnographic Database is a collection of recordings of multiple physiologic signals during sleep. The subjects were monitored in Boston’s Beth Israel Hospital Sleep Laboratory for evaluation of chronic obstructive sleep apnea and to test the effects of constant positive airway pressure (CPAP), a standard therapeutic intervention that usually prevents or substantially reduces airway obstructions in such subjects.

St. Vincent’s University Hospital / University College Dublin Sleep Apnea Database [29]: This contains 25 full overnight polysomnograms with simultaneous three-channel Holter ECG recordings from adult subjects with suspected sleep-disordered breathing.

Haaglanden Medisch Centrum sleep staging database [30]: This is a collection of 151 full-night PSG sleep recordings (85 male, 66 female, mean age: 53.9 ± 15.4 years) collected during 2018 at the Haaglanden Medisch Centrum (HMC, The Hague, The Netherlands) sleep center. The patient recordings were randomly selected and included a heterogeneous population that was referred for PSG examination on the context of different sleep disorders. The dataset contains EEG, EOG, chin EMG, and ECG activity, as well as event annotations corresponding to the scoring of sleep patterns (hypnogram) performed by sleep technicians at HMC. The dataset was collected as part of a study evaluating the generalization performance of an automatic sleep-scoring algorithm across multiple heterogeneous datasets.

EDF-39, EDF-153 [29]: These parts of a public PhysioNetdatabase [31] dataset are often used for benchmarking automatic sleep-stage classification algorithms. As of 2019, the sleep-cassette subset of the database contained 153 full-night polysomnographic sleep recordings of healthy Caucasians aged 25–101 years who did not consume sleep-related medication. We used both the full Sleep-EDF database (EDF-153) and a subset of 39 samples (EDF-39) corresponding to an earlier version of the Sleep-EDF database that has been extensively studied in literature.

Among the above datasets, EDF-39 and EDF-153 are used for experimental accuracy testing, while the others are used for model pre-training. The Fpz-Cz EEG channel was selected for the experiment, and data for 30 s was taken as a cycle; the time before and after going to bed was deleted and not included in the data selection category of this experiment. The s3 and s4 stages were combined into the N3 stage according to the American Academy of Sleep Medicine (AASM) standard.

Simple processing of the sleep signals was required before performing the experiment. During data acquisition, baseline drift occurred in the EEG signals owing to the influence of subject manipulation or other normal physiological activity. To this end, we used median filtering to calibrate the baseline by setting up a temporal neighborhood, and replacing the original data with the median in the neighborhood by taking each set of data as its own neighborhood center. The errors due to baseline drift were greatly reduced by this method. The sleep signals before and after calibration are shown in Figure 4. The sleep-label data were objectively continuous because the time points representsed in the data file were not continuous; thus, the data needed to be completed during processing so that the sleep-label data could be presented as continuous sleep data.

### 3.2. Experimental Result

This section presents our experimental results.The model used *F*1-scores and Accuracy, given the true positives $\left(TP_{i}\right)$, false positives $\left(FP_{i}\right)$, true negatives $\left(TN_{i}\right)$, and false negatives $\left(FN_{i}\right)$ for the *i*-th class, with the Accuracy and *F*1-score defined as follows:(13)$$\begin{array}{c} Accuracy=\frac{\sum_{t=1}^{T}TP_{i}}{N} \end{array}$$(14)$$\begin{array}{c} F1=\frac{1}{T}\sum_{t=1}^{T}\frac{2\times Pre_{t}\times Rec_{t}}{Pre_{t}+Rex_{t}} \end{array}$$
where $Pre_{t}=\frac{TP_{t}}{TP_{t}\times FP_{t}}$; $REC_{t}=\frac{TP_{t}}{TP_{t}\times FN_{t}}$; *T* is the number of classes; and *N* is the total number of samples.

Next, we used SVM [32] and a Gaussian kernel function for sleep staging, Random forest (RF) [33] was an ensemble learning method for sleep staging, and DeepSleepNet [14], SeqSleepNet [34], SleepEEGNet [35], SleepUtime [36], TinySleepNet [37], and SalientSLeepNet [38] were used on EDF39 and EDF153 for comparison. The results were as follows Table 3 and Table 4:

According to the results in Table 3 and Table 4, our model achieved better performance on the sleep-staging task compared to the comparison algorithms. However, on the EDF-39 dataset, our *F*1 score for the N3 stage was relatively lower than that of TinySleepNet, but the other values were higher than those of the other models.

After obtaining the experimental results, we believed that although our model performed classification tasks like the CLIP model, it is doubtful whether the learning between our sleep signals in the CLIP model had improved accuracy with the number of changes. Next, we tested the relationship between model accuracy and amount of pretrained data our model in Figure 5.

From Figure 5, it is clear that as the number of pretraining samples increased, the accuracy improved and approached a steady value. In our initial experiment, we selected accuracy variation results obtained by planning according to different datasets. Then, we randomly selected the same amount of data for pretraining and averaged the accuracy, while still achieving similar results. This experiment confirmed that with increasing sample-size data, the staging accuracy of the model improved accordingly.

### 3.3. Related Experiments

After verifying the accuracy of automatic sleep staging on the two datasets, we conducted more experiments to clarify the direction of improvement of the model.

From Figure 4, we also verified that the dataset size of experimental pretraining would influence final sleep-staging accuracy. It was worth discussing whether the model would have better results if the dataset were increased. Hence, we cross-added the two sets of test data to the pretraining and then cross-tested to see if the model were more accurate with increased data.

From Table 5 and Table 6, we found that the overall accuracy and *F*1 score enabled the model to perform better in the sleep-staging task after increasing the pretraining data. At the same time, we still noted that the *F*1 score of the model did not improve but decreased on some tasks. For example, the *F*1 score of the model in the N2 stage decreased compared to its previous score after adding data participation and training the EDF-153 dataset to the model on the EDF-39 dataset. The reasons for this are further elaborated in the Section 4.

In addition, we tested the influences of different feature extraction methods of sleep signals on the results. Some of these include K-means, SalientSleepNet, and XSleepNet.We mainly test on the EDF-39 dataset, and the results of the test are shown in Table 7.

It is clear that DynamicSleepNet performed the best among these sets of encoders. However, this did not mean that the feature extraction of DynamicSleepNet was the best among all the algorithms. For example, the original purpose of the SalientSleepNet network in the experiment was to extract dual-channel features of EEG and EOG, while the main data in this experiment were EEG single-channelled.

## 4. Discussion

In this part, we mainly propose a feasible improvement direction for the model but it cannot be realized at present owing to the problem of technical capability.

In Table 5 and Table 6, we present the test result of whether the accuracy of the model would continue to improve with increasing pretraining data. The results showed that the overall model result indeed improved, but the score of the model on the classification task for some sleep periods did not improve. We believe that, compared with the task of the CLIP model, the periodicity of the sleep signal itself will cause little difference in the characteristics of the sleep signal between two adjacent sleep periods, leading to poor results in the testing part. To address this issue, we considered data augmentation. We conducted experiments by training with EEG model data augmentation and tested it on the EDF-39 dataset, and the results are as follows:

As shown in Table 8, model performance improved after data augmentation. Unfortunately, this set of experimental data cannot be used as formal experimental data. Because we performed 13 experiments, apart from this one, we obtained reliable experimental data. The remaining experiments resulted in model collapse owing to overfitting problems. The previous experiments were designed and carried out using DeepSleepNet as the encoder. When we changed the encoder to DynamicSleepNet to carry out two experiments, the experimental results still showed model overfitting. After obtaining unique results with DeepSleepNet, we retrained the model five times, and all results showed model overfitting. In conclusion, the results of this experiment are not reproducible and have no reference value.

Although we did not complete a successful experiment, from the existing unreliable experimental results, data augmentation can indeed be considered an improvement direction for this model and other small sample data models based on the CLIP model. As long as the model collapse problem is solved, the accuracy and generalization can be significantly improved with small samples.

## 5. Conclusions

Based on the CLIP model, an automatic sleep-staging model was proposed in this work. To verify model performance, this work selected other automatic staging models to obtain their accuracy on the same EDF39 and EDF153 datasets and compare them. As a result, the model proposed in this work had overall better accuracy on both datasets.

As shown in the experimental section, we found new possibilities and directions for model improvement but did not actually achieve any. The next step, therefore, is to find out why the model was prone to collapse when augmented with data and to devise a solution. With the development of portable medical products owing to the difficulty and inconvenience of EEG signal acquisition, designing sleep-staging models based on other physiological signals that are easy to collect and can maintain high accuracy staging is a direction for the improvement in future models.

## Figures and Tables

**Figure 1 sensors-23-07341-f001:**
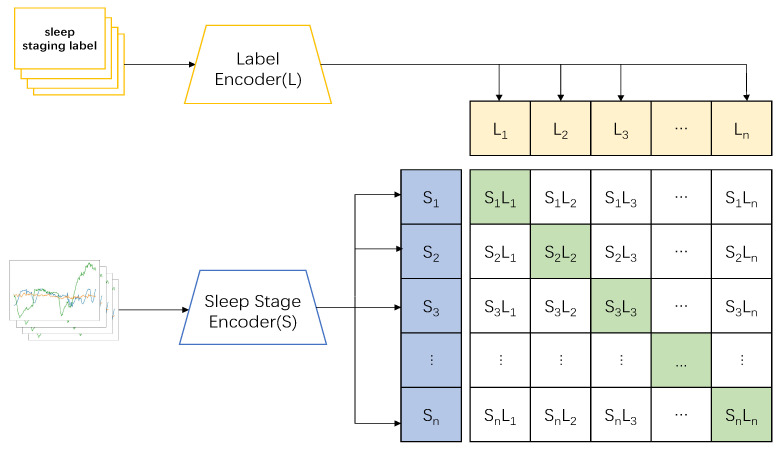
The pre-training model part follows the idea of the CLIP model, which imports the sleep-stage and label signals into the corresponding encoders and outputs the corresponding features (sleep: S, label: L). After that, the model combines the two sets of encoders to predict the correct pairing of a batch of (sleep stage, label) training samples.

**Figure 2 sensors-23-07341-f002:**
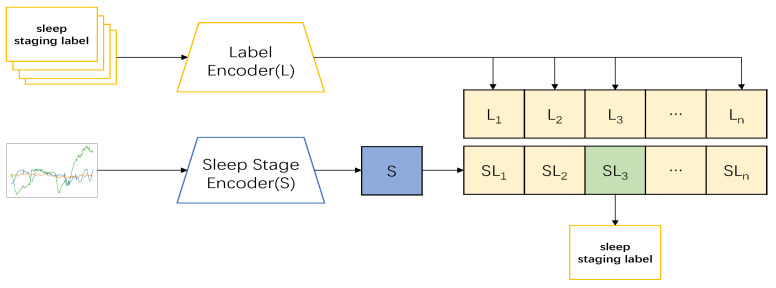
In this part, we input the sleep-staging data of our test into the learned model, let it predict each other with the sleep label data in the model, and took the sleep-label data with the most similar prediction results as the output to obtain the sleep-staging results of the test sleep-staging data.

**Figure 3 sensors-23-07341-f003:**
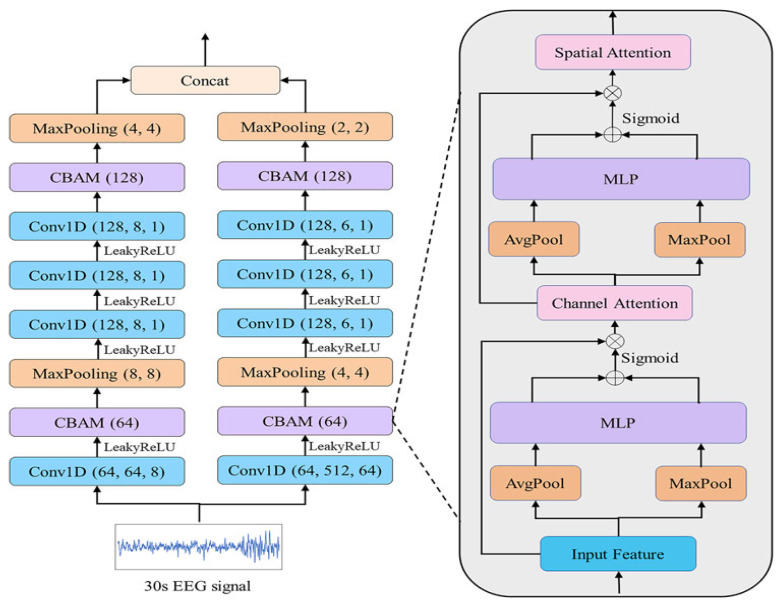
This part is directly borrowed from the EEG feature extraction module of DynamicSleepNet, and the picture was also directly borrowed from the display of this part in the original text of DynamicSleepNet. This module was mainly used to extract effective EEG features, and among these Conv1D (64, 64, 8) refers to using 1D convolution layer with 64 filters, a kernel size of 64 and a stride of 8; Similarly, MaxPooling (8, 8) refers to a maxpooling layer with a kernel size of 8 and a stride of 8; and the leaky rectified linear unit (Leaky–ReLU) refers to the activation function of each convolutional layer.

**Figure 4 sensors-23-07341-f004:**
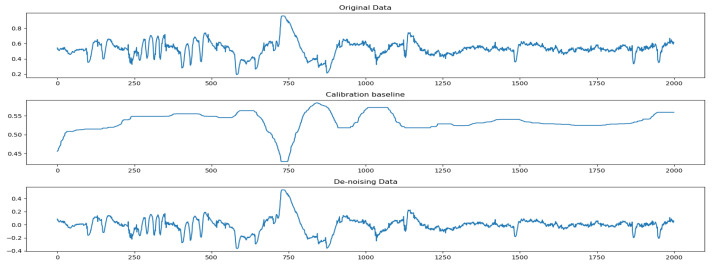
Comparison of noise reduction in EEG signals.

**Figure 5 sensors-23-07341-f005:**
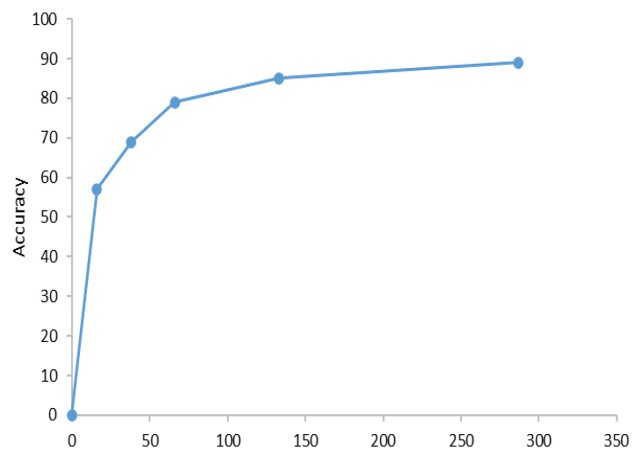
Line plot of the number of people vs. accuracy in pretraining. The abscissa represents the number of pretraining samples, and the ordinate represents the accuracy. (In EDF-153).

**Table 2 sensors-23-07341-t002:** Pseudocode for the overall process of the model.

*CLIP Sleep*	
** *#pre-learning* **	
**1**	input sleep stage (*x*), sleep stage label (*y*)
**2**	*x*, *y* are fed into the encoder to extract features *S*, *L*
**3**	map *S*, *L* to the same vector space
**4**	calculate the cosine similarity between the mapped sleep stages and the labeled sample pairs of sleep stages
**5**	use the obtained cosine similarity values for cross-entropy training
** *#test part* **	
**6**	input test sleep stage ($x_{t}$)
**7**	repeat *x* in steps 2 to 5 to get feature $S_{t}$
**8**	find the $cos<S_{t},L>$ with the highest similarity after training
**9**	output *y*

**Table 3 sensors-23-07341-t003:** Result in EDF-39.

Method	*F*1	Accuracy	W	N1	N2	N3	REM
SVM	63.7	76.1	71.6	13.6	85.1	76.5	71.8
RF	67.6	78.1	74.9	22.5	86.3	80.8	73.3
DeepSleepNet	76.9	82	85	47	86	85	82
SeqSleepNet	79.7	86	91.9	47.8	87.2	85.7	86.2
SleepEEGNet	79.7	84.3	89.2	52.2	86.8	85.1	85
SleepUtime	79	-	87	52	86	85	82
TinySleepNet	80.5	85.4	90.1	51.4	88.5	**88.3**	84.3
SalientSleepNet	83	87.5	92.3	56.2	89.9	87.2	89.2
**Sleep CLIP (ours)**	**84.1**	**89**	**94.2**	**59.7**	**90.2**	87.2	**89.3**

**Table 4 sensors-23-07341-t004:** Result in EDF-153.

Method	*F*1	Accuracy	W	N1	N2	N3	REM
SVM	57.8	71.2	80.3	13.5	79.5	57.1	58.7
RF	62.4	72.7	81.6	23.2	80.6	65.8	60.8
DeepSleepNet	75.3	78.5	91	47	81	69	79
SeqSleepNet	78.2	83.8	92.8	48.9	85.4	78.6	84.1
SleepEEGNet	77	82.8	90.3	44.6	85.7	81.6	82.9
SleepUtime	76	-	92	51	84	78	80
TinySleepNet	78.1	83.1	92.8	51	85.3	81.1	80.3
SalientSleepNet	79.5	84.1	93.3	54.2	85.5	78.3	85.8
**Sleep CLIP (ours)**	**82.9**	**86.9**	**95.3**	**59.4**	**89.1**	**81.7**	**88.9**

**Table 5 sensors-23-07341-t005:** Result in EDF-153 after Adding Data.

Method	*F*1	Accuracy	W	N1	N2	N3	REM
Sleep CLIP (ours)	82.9	87.1	95.3	59.4	89.1	81.7	88.9
Sleep CLIP (EDF-39)	83	87.3	95.4	59.1	89.3	82.1	89.2

**Table 6 sensors-23-07341-t006:** Result in EDF-39 after Adding Data.

Method	*F*1	Accuracy	W	N1	N2	N3	REM
Sleep CLIP (ours)	84.1	89	94.2	59.7	90.2	87.2	89.3
Sleep CLIP (EDF-153)	84.2	89.3	94.3	59.9	89.9	87.5	89.5

**Table 7 sensors-23-07341-t007:** Result in EDF-39 with Different Encoder.

Method	*F*1	Accuracy	W	N1	N2	N3	REM
K-means	51.9	56.1	57.1	29.2	69.9	57.1	46.1
XSleepNet	69	77.4	82.5	34.7	70.2	76.7	81.1
SalientSleepNet	77.47	83.2	92.7	52.5	85.2	73.5	83.2
DeepSleepNet	80.2	85.4	94.5	55.7	87.2	77.9	85.9
DynamicSleepNet	84.1	89	94.2	59.7	90.2	87.2	89.3

**Table 8 sensors-23-07341-t008:** Result in EDF-39 with Augmentation.

Method	*F*1	Accuracy	W	N1	N2	N3	REM
Sleep CLIP (ours)	80.2	85.4	94.5	55.7	87.2	77.9	85.9
Sleep CLIP (Augmentation)	83.5	86.1	94.6	56.7	87.9	78.6	86.2

## Data Availability

All experiments mentioned in this paper use public data sets, and all public data sets are introduced in this paper.

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
