# Peer review of "Sleep CLIP: A Multimodal Sleep Staging Model Based on Sleep Signals and Sleep Staging Labels"

_sensors, 2023, doi:10.3390/s23177341_

Round 1

Reviewer 1 Report

The manuscript cannot be accepted in its present form due to the usage of poor English. There are a few other issues associated with it and hence the authors are advised to revise their manuscript.

1) Please rephrase the title.

2) The introduction section must present the state-of-the-art.

3) Each acronym must be defined when it is used for the first time.

4) There is no discussion from the biosensor's point of view, why?

5) The accuracy is still not very good, what steps will be taken to improve the accuracy beyond 90%?

The English language in this manuscript is of poor quality. The authors must revise their manuscript to remove grammatical errors.

Reviewer 2 Report

The manuscript describes designing a multimodal model based on the CLIP model, which utilises sleep signals and labels for sleep staging. The authors pre-trained the model using five different training sets and then tested their proposed method on two specific training sets, namely EDF-39 and EDF-153. The proposed approach to sleep staging using the CLIP model demonstrates some results in achieving high accuracy in diagnosing sleep disorders based on two different training sets. The manuscript is well-explained; however, the following comments need to be considered for more clarification.

1-    Many abbreviations were used in the manuscript without referring to the full name. All full names need to be referred to at least once as this research, as non-experts may study it.

2-    In the Materials and Methods section: The authors claimed that “the model uses a large amount of Internet data for pre-training”. Can the authors pre-train the model by a specific dataset, e.g. a dataset for EEG? Is it possible to predefine the required information before any modelling? Please elaborate. More clarification is required.

3- According to the authors, other modelling can be used in such studies. What are of advantages and disadvantages of using the CLIP model for this research?

4- What sort of filtering was used in this study? What was the reason for choosing this method of filtering? Please elaborate.

5- The parameters of the tables should be accurately explained.

6- In the Conclusion section, what are the criteria of a “good performance” on the proposed model in this study?

Round 2

Reviewer 1 Report

The revised manuscript is now in better shape to be accepted for publication.

Reviewer 2 Report

The revised manuscript is now in better shape to be accepted for publication.

The English language in this manuscript is of poor quality. The authors must revise their manuscript to remove grammatical errors.